# Melatonin Attenuates H_2_O_2_-Induced Oxidative Injury by Upregulating LncRNA NEAT1 in HT22 Hippocampal Cells

**DOI:** 10.3390/ijms232112891

**Published:** 2022-10-25

**Authors:** Qiang Gao, Chi Zhang, Jiaxin Li, Han Xu, Xiaocheng Guo, Qi Guo, Chen Zhao, Haixu Yao, Yuhan Jia, Hui Zhu

**Affiliations:** Department of Physiology, Harbin Medical University, Harbin 150081, China

**Keywords:** melatonin, LncRNA, NEAT1, oxidative injury, *Slc38a2*

## Abstract

More research is required to understand how melatonin protects neurons. The study aimed to find out if and how long non-coding RNA (lncRNA) contributes to melatonin’s ability to defend the hippocampus from H_2_O_2_-induced oxidative injury. LncRNAs related to oxidative injury were predicted by bioinformatics methods. Mouse hippocampus-derived neuronal HT22 cells were treated with H_2_O_2_ with or without melatonin. Viability and apoptosis were detected by Cell Counting Kit-8 and Hoechst33258. RNA and protein levels were measured by quantitative real-time PCR, Western blot, and immunofluorescence. Bioinformatics predicted that 38 lncRNAs were associated with oxidative injury in mouse neurons. LncRNA nuclear paraspeckle assembly transcript 1 (NEAT1) was related to H_2_O_2_-induced oxidative injury and up-regulated by melatonin in HT22 cells. The knockdown of NEAT1 exacerbated H_2_O_2_-induced oxidative injury, weakened the moderating effect of melatonin, and abolished the increasing effect of melatonin on the mRNA and protein level of *Slc38a2*. Taken together, melatonin attenuates H_2_O_2_-induced oxidative injury by upregulating lncRNA NEAT1, which is essential for melatonin stabilizing the mRNA and protein level of *Slc38a2* for the survival of HT22 cells. The research may assist in the treatment of oxidative injury-induced hippocampal degeneration associated with aging using melatonin and its target lncRNA NEAT1.

## 1. Introduction

The hippocampus is in charge of emotional regulation, spatial learning, and memory, all of which are important in people’s studies and daily lives. Neurons in the hippocampus undergo senescence and apoptosis as they age, impairing cognition and leading to dementia such as Alzheimer’s disease [1]. However, it is unclear which approaches and medications can effectively prevent hippocampus aging.

Oxidative injury caused by H_2_O_2_ is one of the leading causes of hippocampal aging [2]. In the central nervous system, H_2_O_2_ mainly comes from neurons during the metabolic process [3] and accumulates with aging. In turn, the accumulated H_2_O_2_ degrades the quality of proteins, RNA, DNA, cell membranes, and mitochondria, hastening neuron aging and even inducing death [4]. However, it is still difficult to prevent or reduce the chronic injury induced by H_2_O_2_ during hippocampal aging.

Long non-coding RNAs (lncRNAs) are found in a wide range of tissues, including brain neurons and glial cells. They influence neural activity and disease progression. In terms of oxidative injury, Junsang et al. discovered that the overexpression of lncRNA nuclear paraspeckle assembly transcript 1 (NEAT1) resisted H_2_O_2_-induced oxidative injury in mouse neuroma Neuro2A cells [5]. Ran et al. found that knocking out of lncRNA ANRIL enhanced H_2_O_2_-induced oxidative injury by increasing microRNA-125a expression in rat PC12 nerve cell lines [6]. Ghattas et al. reported that lncRNA MALAT1 promoted the survival of HT22 cells in the presence of H_2_O_2_ [7]. However, there is still a lack of research on the resistance of lncRNA to H_2_O_2_-induced oxidative injury in hippocampus neurons, which leads to a superficial understanding of the role and mechanism of lncRNA against H_2_O_2_-induced oxidative injury.

Melatonin is an amine hormone secreted by the pineal gland and in mammals. In addition to a variety of beneficial functions for the body, melatonin can safeguard nerve system. Melatonin resists the damage of H_2_O_2_ to neural stem cells through the melatonin receptor-mediated PI3K/AKT pathway [8]. Melatonin protects against prions’ neurotoxicity [9]. Melatonin has also been demonstrated to help with the symptoms of Alzheimer’s disease and other neurodegenerative ailments [10]. Our previous study found that melatonin did not affect the morphology and activity of HT22 cells, but it inhibited H_2_O_2_-induced oxidative injury through activating autophagy [11]. However, relatively few research studies have been conducted to determine whether and how melatonin protects hippocampal neurons from H_2_O_2_-induced oxidative injury.

Studies have shown that melatonin protects cells by regulating lncRNA. Cai et al. found that melatonin prevents senescence and apoptosis in cardiac progenitor cells induced by H_2_O_2_ via upregulating lncRNA H19 [12]. Song et al. reported that melatonin upregulated lncRNA H19 expression, which resisted the senescence and apoptosis of glioma cells caused by H_2_O_2_ [13]. However, it is yet uncertain if melatonin affects the expression of lncRNA in the hippocampus and whether any other lncRNA mediates melatonin’s inhibition of H_2_O_2_-induced oxidative injury.

In this study, several lncRNAs, including NEAT1, 1810026B05Rik, and small nucleolar RNA host gene 12 (SNHG12), were predicted to be related to oxidative injury in mouse neurons by bioinformatic analysis. HT22 cells were treated with H_2_O_2_ to induce chronic oxidative injury and thereby stimulate the occurrence of senescence. HT22 cells were also administered with melatonin to observe the resistance of melatonin to H_2_O_2_-induced oxidative injury and the expression of lncRNA (NEAT1, 1810026B05Rik, SNHG12). To further confirm the role and mechanism of melatonin, plasmids expressing hairpin RNA to silence the expression of lncRNA NEAT1 were transfected into HT22 cells. The study sought to demonstrate that melatonin protects hippocampal neurons from H_2_O_2_-induced oxidative injury by upregulating lncRNA NEAT1.

## 2. Results

### 2.1. LncRNA and mRNA Expression Profile of Oxidative Injury in Mouse Neurons

GSE22087 microarray data related to oxidative injury in neurons were obtained from the GEO database. The Affymetrix Gene Chip Mouse Genome 430 2.0 chip, which contained 45,057 probes and identified more than 39,000 transcripts, was used in the microarray study. According to the probe annotation file’s relevant gene names and the mouse GENCODE m38.p6, 38,300 specific probes were obtained, including 1800 lncRNA-related probes and 31,929 mRNA-related probes (Table 1). LncRNA and mRNA expression profile data were generated, which contained 1439 lncRNAs and 15,965 mRNAs.

### 2.2. Differential Expression Analysis of lncRNA and mRNA

To analyze the differential expression of lncRNA and mRNA in the process of oxidative injury, the expression distributions of lncRNA and mRNA were firstly observed by the scatter plot. As shown in Figure 1A,B, the distribution of lncRNA and mRNA differentially expressed in the oxidative injury group and control group were screened out. The horizontal axis represents the value of lncRNA and mRNA in the control group, while the vertical axis represents the value of lncRNA and mRNA in the oxidative injury group. The red dots represent lncRNA or mRNA in each sample. The lncRNA or mRNA distribution along the intermediate line (x = y) represents a similar expression in both groups. The dots outside of the two edges (|y − x| > 1) reflect lncRNA or mRNA that differ by more than a 2-fold difference between the two groups.

To further identify the differentially expressed lncRNA and mRNA, the *t*-test model of SAM q < 0.1 was applied. When q was set < 0.1, a total of 38 differentially expressed lncRNAs were identified, all of which were upregulated; a total of 1193 mRNAs were differentially expressed, 773 of which were upregulated and 420 of which were downregulated. The differential expressed lncRNAs and mRNAs were exhibited by heat maps (Figure 1C,D), with all of the differentially expressed lncRNAs and the top 50 differentially expressed mRNAs listed. The differently expressed lncRNA and mRNA were also listed in Table 2 and Table 3, respectively.

### 2.3. Homology Analysis of Differentially Expressed lncRNA

To increase the possibility of transformation and the practical value of lncRNA, the 38 differentially expressed lncRNAs are evaluated by the homology analysis between the human and the mouse transcriptomes. The analysis screened out three lncRNAs with high homology between humans and mice, including NEAT1, 1810026B05Rik, and 2900009J06Rik. The analysis also screened out three lncRNAs with medium homology, including SNHG12, SNHG1, and C130026L21Rik (Table 4). More genomic information of these six lncRNAs is listed in Table 5.

Pearson correlation analysis was performed based on the six differentially expressed lncRNAs and the 1193 differentially expressed mRNAs. For each lncRNA, mRNA with a Pearson correlation coefficient > 0.5 and a *p* value < 0.01 were considered to be significantly co-expressed with the lncRNA. As a result, 806 mRNAs were identified to be significantly associated with the six lncRNAs, resulting in 2701 lncRNA-mRNA correlation pairs, as illustrated in Appendix A. The triangle dots represent lncRNA. The round dots represent lncRNA-associated mRNA, while the red round dots are mRNA that have been related to aging, apoptosis, proliferation, or autophagy. According to the findings, there were 556 SNHG1-associated mRNAs, 629 SNHG12-associated mRNAs, 95 NEAT1-associated mRNAs, 256 1810026B05Rik-associated mRNAs, 585 2900009J06Rik-associated mRNAs, and 580 C130026L21Rik-associated mRNAs.

### 2.4. LncRNA NEAT1 and 1810026B05Rik Were Involved in Melatonin Protecting against H_2_O_2_-Induced Oxidative Injury in HT22 Cells

To confirm lncRNA involvement in oxidative injury in neurons, HT22 cells were treated with 200 μM H_2_O_2_ for 24 h. LncRNA expression levels of the six lncRNAs (NEAT1, 1810026B05Rik, 2900009J06Rik, SNHG12, SNHG1, and C130026L21Rik) were detected by quantitative real-time PCR (qPCR). As shown in Figure 2A, lncRNA expressions of NEAT1, SNHG12, and 1810026B05Rik in the H_2_O_2_ group increased significantly, which was consistent with prior results of bioinformatic prediction. However, there was no significant difference in the expressions of 2900009J06Rik, SNHG1, and C130026L21Rik between the two groups. Our previous research has proved that melatonin protects HT22 cells from H_2_O_2_-induced oxidative injury. Hence, to further identify whether lncRNA is involved in melatonin protecting against H_2_O_2_-induced oxidative injury, HT22 cells were treated with 200 μM H_2_O_2_ or 200 μM H_2_O_2_ + 50 μM melatonin. LncRNA expressions of NEAT1, 1810026B05Rik, and SNHG12 were detected by qPCR. As demonstrated in Figure 2B, melatonin effectively suppressed the upregulated expressions of NEAT1, SNHG12, and 1810026B05Rik caused by H_2_O_2_. This indicates that the expression of NEAT1, SNHG12, and 1810026B05Rik can be raised by H_2_O_2_, while melatonin suppresses the effect of H_2_O_2_.

To further investigate the influence of melatonin on the expression of NEAT1, SNHG12, and 1810026B05Rik, HT22 cells were treated with 50 μM melatonin for 24 h, and then, the expression of NEAT1, SNHG12, and 1810026B05Rik was detected by qPCR. As shown in Figure 2C, melatonin significantly upregulated the expression of NEAT1 and 1810026B05Rik but had no significant effect on SNHG12. This suggested that NEAT1 and 1810026B05Rik were probably involved in melatonin’s ability to prevent H_2_O_2_-induced oxidative injury. As a result, we decided to explore the potential role of NEAT1 in melatonin’s rescue of H_2_O_2_-induced oxidative injury.

### 2.5. Gene Ontology Analysis for the Potential Functionalities of NEAT1

Biological processes (BP) of Gene Ontology (GO) functional enrichment analysis were performed based on the above correlation analysis of the six lncRNAs and their 806 co-expressed mRNAs.

As shown in Figure 3A,B, GO BP analysis of NEAT1 was conducted on its 95 co-expressed mRNAs, which were considerably enriched into 28 BP terms. Among the 28 BP terms, there were BPs significantly associated with proliferation, apoptosis, senescence, and autophagy, such as endoplasmic reticulum (ER) overload response, negative regulation of cell proliferation, cell cycle arrest, negative regulation of the apoptotic process, positive regulation of apoptotic process and cellular response to glucose starvation.

Those bioinformatic results predicted that NEAT1 was significantly related to proliferation, apoptosis, senescence, and autophagy.

### 2.6. The Knockdown of lncRNA NEAT1 Aggravated H_2_O_2_-Induced Oxidative Injury in HT22 Cells

To further confirm the functions and effects of NEAT1 in H_2_O_2_-induced oxidative injury, short hairpin RNA-negative control (shRNA-NC), and shRNA-NEAT1 expression vectors were transfected into HT22 cells for 48 h. As illustrated in Figure 4A, cells were transfected with vectors expressed green fluorescent protein (GFP), which can be observed under a fluorescence microscope. As shown in Figure 4B, shRNA-NC had no effect on NEAT1 expression when compared to the control group. ShRNA-NEAT1 plasmid decreased NEAT1 expression when compared to the shRNA-NC group.

After the transfection with shRNA-NC or shRNA-NEAT1 expression vectors for 48 h, HT22 cells were treated with H_2_O_2_ for 24 h. Cell morphology, vitality, and apoptosis were then observed. As seen in Figure 4C–F, the cell morphology, viability, and apoptosis were not obviously affected by shRNA-NC when compared to the H_2_O_2_ group. In comparison to the H_2_O_2_ + shRNA-NC group, shRNA-NEAT1 worsens the injury effects of H_2_O_2_ on cellular morphology, viability, and apoptosis.

Those results indicated that the knockdown of NEAT1 exacerbated H_2_O_2_-induced oxidative injury in HT22 cells, implying that NEAT1 is necessary for HT22 cell survival in the face of H_2_O_2_-induced oxidative injury.

### 2.7. The Knockdown of lncRNA NEAT1 Inhibited the Protective Effect of Melatonin on H_2_O_2_-Induced Oxidative Injury

To further confirm that NEAT1 is required for melatonin in protecting against H_2_O_2_-induced oxidative injury, HT22 cells were transfected with shRNA-NEAT1 expression vectors for 48 h followed by treatment with H_2_O_2_ or H_2_O_2_ + melatonin for 24 h.

As shown in Figure 5A, melatonin significantly inhibited H_2_O_2_-induced cellular swelling and death, which was effectively abolished by shRNA-NEAT1. As seen in Figure 5B, melatonin obviously prevented H_2_O_2_-induced the decrease in cellular viability (*p* < 0.05), which was significantly attenuated by shRNA-NEAT1 (*p* < 0.05). As shown in Figure 5C,D, apoptosis was greatly decreased in the H_2_O_2_ + melatonin group as compared to the H_2_O_2_ group. However, when compared to the H_2_O_2_ + melatonin group, apoptosis was obviously in the H_2_O_2_ + melatonin + shRNA-NEAT1 group.

Those findings revealed that NEAT1 was critical for melatonin’s resistance to H_2_O_2_-induced oxidative injury.

### 2.8. The Knockdown of NEAT1 Disturbed mRNA Level of Eif2ak3 and Slc38a2

To figure out the underlying reason why the knockdown of NEAT1 weakens the effect of melatonin, the influence of H_2_O_2_, melatonin, and shRNA-NEAT1 on the expressions of NEAT1 co-expressed mRNA was further detected. HT22 cells were transfected with or without shRNA-NC or shRNA-NEAT1 vector for 48 h, which was followed by treatment with melatonin, H_2_O_2,_ or H_2_O_2_ + melatonin for 24 h. The level of NEAT1 co-expressed mRNA *Atf4*, *Dap*, *Eif2ak3*, *Gata1*, *Ifn-γ*, and *Slc38a2* was then detected by qPCR.

As demonstrated in Figure 6A, melatonin or H_2_O_2_ significantly upregulated *Atf4* in comparison to the control group; H_2_O_2_ + melatonin downregulated *Atf4* in comparison to the H_2_O_2_ group; and shRNA-NEAT1 did not affect *Atf4* in comparison to the H_2_O_2_ + melatonin + shRNA-NC group. Dap did not differ substantially between groups (Figure 6B). As shown in Figure 6C, *Eif2ak3* decreased in the H_2_O_2_ + melatonin group compared to the H_2_O_2_ group but increased in the H_2_O_2_ + melatonin + shRNA-NEAT1 group compared to the H_2_O_2_ + melatonin + shRNA-NC group. As seen in Figure 6D, melatonin downregulated *Gata1* when compared to the control group; H_2_O_2_ elevated *Gata1*, which was significantly suppressed by melatonin; and shRNA-NEAT1 did not affect *Gata1* when compared to the H_2_O_2_ + melatonin + shRNA-NC group. H_2_O_2_ raised the level of *Ifn-γ* mRNA, which was unaffected by other treatments (Figure 6E). As compared to the control group, *Slc38a2* was decreased by H_2_O_2_, which was dramatically reversed by melatonin (Figure 6F); as compared to the H_2_O_2_ + melatonin + shRNA-NC group, *Slc38a2* was significantly decreased by shRNA-NEAT1 (Figure 6F).

Those results revealed that the knockdown of NEAT1 diminished the effect of melatonin on the mRNA levels of *Eif2ak3* and *Slc38a2*.

### 2.9. The Knockdown of NEAT1 Inhibited Protein Level of Slc38a2

Sodium-coupled neutral amino acid transporter 2 (SLC38A2) is a transporter of short-chain amino acids in cell membranes, which plays an important role in cell survival and neural function. Therefore, the influence of H_2_O_2_, melatonin, and shRNA-NEAT1 on the expression of SLC38A2 was further detected. Followed by treatment with melatonin, H_2_O_2,_ or H_2_O_2_ + melatonin for 24 h, HT22 cells were transfected with or without shRNA-NC or shRNA-NEAT1 vector for 48 h. Then, SLC38A2 was detected by Western blot or immunofluorescence analysis.

Western blot analysis showed that SLC38A2 was greatly reduced by H_2_O_2_ compared to the control group, which was significantly reversed by melatonin (Figure 7A). When compared to the H_2_O_2_ + melatonin + shRNA-NC group, SLC38A2 was considerably decreased in the H_2_O_2_ + melatonin + shRNA-NEAT1 group (Figure 7A). As shown in Figure 7B, immunofluorescence staining revealed a comparable result to the Western blot analysis.

Those results suggested that the knockdown of NEAT1 hindered the protective effect of melatonin by inhibiting the protein level of *Slc38a2.*

## 3. Discussion

In this study, we analyzed a set of microarray data (GSE22087) of oxidative injury in mouse neurons. After probe reannotation, a set of RNA expression profile data containing 1439 lncRNAs and 15,965 mRNAs was obtained. To investigate lncRNA and mRNA significantly associated with oxidative injury in neurons, 38 differentially expressed lncRNAs and 1193 differentially expressed mRNAs were found out by differential expression analysis. Despite several studies on lncRNA in recent years, only a handful of lncRNAs have formal names and defined functions. Therefore, the majority of the 38 lncRNAs’ functions were blank.

As observed in many studies, most lncRNAs are not as conserved as mRNA among species [14]. Even if they have the same name, the sequences of lncRNAs from different species, including mice and humans, could be different. This characteristic may impede research into the functions of non-conserved lncRNA from mice to humans. However, there are some lncRNAs with conserved sequences, which are worth studying in mice to reflect possible functions in humans. Therefore, lncRNAs with high homology between humans and mice were predicted to increase the possibility of transformation and the practical value for lncRNA. Current homology prediction methods, mainly based on alignment algorithms such as BLAST, assume the equivalence between homology and nucleotide sequence similarity [15]. By applying the BLAST database, six lncRNAs with a high or medium level of homology between mice and humans were screened out from the 38 differentially expressed lncRNAs. The six lncRNAs are NEAT1, 1810026B05Rik, 2900009J06Rik, SNHG12, SNHG1, and C130026L21Rik.

Of the six lncRNAs, the functions of NEAT1, SNHG12, and SNHG1 have been reported in cancer as well as neuron injury. However, research on 1810026B05Rik, 2900009J06Rik, and C130026L21Rik is currently inadequate. The co-expression network is frequently used to represent gene regulation, which is the functional annotation of unknown genes [16]. Based on the Pearson correlation analysis, 806 differentially expressed mRNAs related to the six lncRNAs were identified, and the lncRNA–mRNA co-expression network concerning oxidative injury was constructed. There are mRNA-regulating biological processes of aging, apoptosis, proliferation, and autophagy in the network, which are reported to participate in oxidative injury. As a result, the findings suggested that the six lncRNAs are likely to be involved in oxidative injury. NEAT1, SNHG12, and SNHG1 have been related to oxidative stress injury in neurons [5,17,18], which validates our prediction.

H_2_O_2_ is widely used to induce oxidative injury [12], and lncRNAs have been implicated in H_2_O_2_-induced oxidative injury in neurons. For example, lncRNA H19 was decreased by H_2_O_2_ in U87 cells, while melatonin protected the cells by upregulating H19 [13]. Epigenetically-induced lncRNA1 (EPIC1) was downregulated in H_2_O_2_-treated SH-SY5Y neuronal cells and primary human neurons, while the overexpression of lncRNA EPIC1 significantly attenuated H_2_O_2_-induced death and apoptosis in neurons [19]. To further confirm whether the six lncRNAs are involved in H_2_O_2_-induced oxidative injury in HT22 cells, their expression was detected following the H_2_O_2_ treatment. The result showed that NEAT1, SNHG12, and 1810026B05Rik were upregulated by H_2_O_2_, which was consistent with our previous prediction. However, SNHG1, 2900009J06Rik, and C130026L21Rik were not affected by H_2_O_2_, suggesting the three lncRNAs were not related to H_2_O_2_-induced oxidative injury.

Melatonin has been widely demonstrated to reduce H_2_O_2_-induced oxidative injury through various mechanisms. We previously found that melatonin inhibited H_2_O_2_-induced oxidative injury in HT22 cells by upregulating BECLIN1-autophagy-related genes (ATGs) and activating autophagy [11]. Moreover, lncRNAs, such as H19 and maternally expressed gene 3 (MEG3), were discovered to mediate the neural protection of melatonin against the H_2_O_2_-induced oxidative injury [13,20]. Therefore, to figure out whether NEAT1, SNHG12, and 1810026B05Rik were associated with melatonin’s protection against H_2_O_2_-induced oxidative injury, the three lncRNAs were detected after the treatment of melatonin in the presence or absence of H_2_O_2_. The result showed that melatonin prevented the upregulation of the three lncRNAs induced by H_2_O_2_. Interestingly, melatonin did not downregulate the three lncRNAs but did upregulate NEAT1 and 1810026B05Rik in the absence of H_2_O_2_. As a result, we hypothesized that the suppressing effect of melatonin on the upregulation of NEAT1, SNHG12, and 1810026B05Rik induced by H_2_O_2_ was not directly generated by melatonin regulation but rather because melatonin weakened the stimulating effect of H_2_O_2_ on these lncRNAs. The results also suggested that NEAT1 and 1810026B05Rik were more likely implicated in melatonin neuron protection against H_2_O_2_. Therefore, we further concentrated our efforts in this study on investigating the potential functions of NEAT1.

To further identify the functions of lncRNA NEAT1, gene ontology for their potential functionalities was analyzed and presented in the form of networks. NEAT1 was found to be highly related to the biological processes of proliferation, apoptosis, senescence, autophagy, and oxidative stress. These biological processes are all linked to oxidative injury, implying the functions of NEAT1 warrant further investigation. Moreover, of the 95 NEAT1-associated mRNAs, *Npas4*, *Taf15*, *Lgals3*, *Il1a*, *Ifng*, *Arl4d*, and *Arhgap26* have been related to NEAT1 in previous research [21,22,23,24,25]. Therefore, these mRNAs and their corresponding proteins should be noted during discussing functions of NEAT1 in future research.

Furthermore, NEAT1 in HT22 cells was knocked down to observe the subsequent effects on H_2_O_2_-induced oxidative injury and melatonin protection. We found that the knockdown of NEAT1 exacerbated the injury effect of H_2_O_2_, indicating that NEAT1 is required for HT22 cell survival in oxidative injury. Several types of research showed that NEAT1 prevented neuron injury. Sunwoo et al. discovered that NEAT1 was significantly upregulated in the brain of Huntington mice, and that the overexpression of NEAT1 in the neuro2A cell line of mice could withstand H_2_O_2_-induced damage and improve cell survival [5]. Zhong et al. found that ischemia and hypoxic injury upregulated NEAT1 expression in primary mouse neurons, while the knockdown of NEAT1 inhibited neuronal axon elongation [26]. Liu et al. observed that NEAT1 increased in the cerebral cortex surrounding the craniocerebral trauma site, promoting the recovery of brain function in mice. On the other hand, NEAT1 knockdown hindered nerve recovery following injury [27]. Therefore, we concluded that the increased expression of NEAT1 in the presence of H_2_O_2_ was an intracellular triggered response to H_2_O_2_ with the goal of alleviating the injury produced by H_2_O_2_.

Up to now, there have been no reports on the link between NEAT1 and melatonin. In this study, the knockdown of NEAT1 weakened melatonin’s resistance to H_2_O_2_-induced oxidative injury in HT22 cells. The result revealed that NEAT1 is essential for melatonin to counteract H_2_O_2_-induced neuron injury.

Why did the knockdown of NEAT1 inhibit the action of melatonin? To figure out the question, the influence of H_2_O_2_, melatonin, and shRNA–NEAT1 on the expressions of lncRNA co-expressed mRNAs was detected. In the presence of H_2_O_2_ + melatonin, the knockdown of NEAT1 significantly disrupted the mRNA levels of *Eif2ak3* and *Slc38a2* compared to the H_2_O_2_ + melatonin + shRNA-NC group.

*Eif2ak3*, also known as protein kinase r-like endoplasmic reticulum kinase (PERK), is a signaling pathway protein activated by ER stress. PERK phosphorylates and activates eukaryotic initiation factor 2 (eIF2) as p-eIF2, which mediates the transient and rapid inhibition of translation to reduce the formation of misfolded proteins [28]. Increased p-eIF2 also impairs memory formation [29], and the upregulation and overactivation of PERK are associated with neurodegeneration and memory impairment [30,31]. Sharma et al. found that eliminating PERK from the hippocampus of mice prevented hippocampal senescence and memory decline [32]. In addition, the activation of PERK inhibited cyclin D1 mRNA, thus inhibiting cell proliferation [33]. In this study, *Eif2ak3* mRNA was decreased in the H_2_O_2_ + melatonin group compared to the H_2_O_2_ group, and it increased in the H_2_O_2_ + melatonin + shRNA-NEAT1 group compared to the H_2_O_2_ + melatonin + shRNA-NC group. Therefore, we suppose that the knockdown of NEAT1 increases PERK, which suppresses cell protein translation through p-eIF2, thus inhibiting the protective effect of melatonin and exacerbating cell damage.

SLC38A2, also known as SNAT2, is a transporter of short-chain amino acids in cell membranes, which mainly transports alanine, serine, proline, and glutamine. SLC38A2 expression is regulated by various cell stress responses, such as endoplasmic reticulum stress and neutral amino acids depletion [34,35]. Jeon et al. found that inhibiting SLC38A2 expression under endoplasmic reticulum stress could limit the mTOR signaling pathway and cell proliferation by reducing glutamine uptake, ultimately leading to apoptosis [36]. Duval et al. discovered that adiponectin reduces the expression of SLC38A2 and promotes cell death in placental cells [37]. Thus, SLC38A2 plays a key role in cell function and survival. Our result showed that mRNA and protein levels of *Slc38a2* increased in the H_2_O_2_ + melatonin group compared to the H_2_O_2_ group, but they decreased in the H_2_O_2_ + melatonin + shRNA-NEAT1 group compared to the H_2_O_2_ + melatonin + shRNA-NC group. As a result, we conclude that melatonin inhibits H_2_O_2_-induced oxidative injury through upregulating *Slc38a2* expression, which is necessary for amino acid transport. By decreasing *Slc38a2* expression, the knockdown of NEAT1 weakens the protective effect of melatonin and leads to apoptosis.

In this study, H_2_O_2_ increased *Atf4* and *Gata1* mRNA, which were inhibited by melatonin.

Activating transcription factor 4 (ATF4) is a well-known pressure-induced transcription factor of the ATF/CREB family, which regulates the expression of growth factors, cytokines, chemokines, and adhesion molecules [38]. *Atf4* was found to be upregulated in many pathological states, including H_2_O_2_-induced oxidative injury [39], which is consistent with our result. Gully et al. discovered that the overexpression of *Atf4* activated caspases 3/7, which led to severe degeneration and apoptosis of neurons in the striatum nigra of mice [40]. Galehdar et al. found that the overexpression of *Atf4* caused mouse cortical neurons more sensitive to stress-induced apoptosis, while the knockdown of *Atf4* significantly reduced neuron death by lowering P53 up-regulated modulator of apoptosis (PUMA) [41]. Therefore, we suppose that the rise in *Atf4* induced by H_2_O_2_ promotes apoptosis and makes cells more susceptible to H_2_O_2_-induced oxidative injury, and that melatonin decreases *Atf4* to relieve the H_2_O_2_-induced oxidative injury. Interestingly, we also found that melatonin upregulated *Atf4* in the absence of H_2_O_2_. It is also known that *Atf4* upregulates the C/EBP homologous protein (CHOP) and *Map1lc3B* to stimulate the expression of autophagy-related genes [42,43]. Our previous work also showed that 50 μM melatonin upregulated autophagy by increasing the ratio of LC3B 14/16kd [11]. Those results suggest that upregulating *Atf4* might be another mechanism of melatonin activating autophagy in HT22 cells.

There is currently no study to prove the expression and function of *Gata1* in neurons. GATA binding protein 1 (GATA1) is an important transcription factor in bone marrow hematopoietic cells, which can promote the expression of multiple genes related to proliferation, differentiation, and survival in hematopoietic stem cells [44]. The abnormal expression of *Gata1* is closely related to a variety of hematopoietic diseases. Therefore, we speculate that the abnormal upregulation of *Gata1* caused by H_2_O_2_ is related to the injury in HT22 cells, and that melatonin restores *Gata1* to normal levels to relieve H_2_O_2_-induced oxidative injury.

Some lncRNAs have been identified as biomarkers for a variety of diseases. For example, Li et al. found that the quantity of H19 in the liver of mice and humans was positively linked with the degree of liver damage caused by cholestasis [45]. Huang et al. found that the level of lncRNA X-inactive Specific Transcript (XIST) in urine was proportional to the degree of glomerular injury in mice [46]. We also observed that H_2_O_2_ upregulated the expression of lncRNA SNHG12, whereas melatonin downregulated SNHG12 while resisting H_2_O_2_-induced oxidative injury. However, melatonin did not affect SNHG12 expression under physiological conditions. This indicates that the expression level of SNHG12 may be an indicator of oxidative injury. Hence, the expression of SNHG12 increases with neuron injury induced by H_2_O_2_. However, as melatonin reduces the injury caused by H_2_O_2_, its expression decreases.

## 4. Materials and Methods

### 4.1. RNA Microarray Data of Oxidative Injury in Mouse Neurons

To study the role of lncRNA during oxidative injury in neurons, a set of RNA microarray data (NO. GSE22087) and its relevant probe annotation file were obtained from the GEO database. The RNA for the microarray data was extracted from primary cultured cortical neurons of mice. The oxidative injury model was established by nitric oxide treatment for 24 h. There are 3 samples of RNA in the oxidative injury group and 5 samples in the normal control group.

All transcript names provided in the probe annotation file of GSE22087 were annotated to screen mRNA probes and lncRNA probes. Firstly, the probe annotation file was sorted out. In the annotation file, if a probe corresponds to multiple transcripts, the unspecific probe will be removed from our data analysis experiment. Then, m38.p6 annotation data of mouse genome and lncRNA annotation data were downloaded from the GENCODE database. Then, mRNA names were extracted from the annotated data of the whole mouse genome, which will be matched with the mRNA names in the annotated file of GSE22087. Finally, the probes for detecting mRNA in the annotated file were screened out. With a similar method, the probes for detecting lncRNA in the annotated file were also screened out.

### 4.2. Construction of lncRNA and mRNA Expression Profiles

With the obtained lncRNA probes and mRNA probes, the expression values of the probes in each sample were assigned to corresponding genes, and then, the expression profile data of lncRNA and mRNA were constructed. If multiple probes detected the same gene, the median algorithm was adopted to calculate the expression value of the gene.

The expression trends of lncRNA and mRNA were demonstrated by scatter plots. Firstly, the mean values of expression values of lncRNA and mRNA were calculated and converted into log2. Using the R package “ggplot2”, set the horizontal axis for each gene expression in the control group, the vertical axis for each gene expression in the oxidative injury group, and the observation line y = x and |y − x| > 1. When |y − x| > 1, the fold change (FC) > 2 or < 0.5. FC > 2 was defined as upregulated genes and FC < 0.5 as downregulated genes.

### 4.3. Cluster Analysis of Differential Expression

According to the lncRNA and mRNA expression profiles previously constructed, the SAM method (q < 0.1) was used for differential expression analysis, which was completed by the SAM function in “samr” of the R package. To visualize the results of the analysis, a cluster analysis was presented using the “Pheatmap” package in the R language.

### 4.4. Homology Analysis

The complete sequence information of mouse lncRNA was obtained from the Ensemble database, and then, the sequence information was compared to the human transcriptome in the BLAST database of the National Center for Biotechnology Information (NCBI). The results showed whether and how the mouse lncRNA was homologous with human RNA. A score of the degree of homology greater than 200 was considered as high homology, a score between 100 and 200 was medium homology, and a score less than 100 low homology.

### 4.5. Pearson Correlation Analysis

The potential interaction between lncRNA and mRNA can be realized by finding the co-expression patterns of lncRNA and mRNA. Among the existing methods based on expression data, Pearson correlation analysis is the main method to identify co-expressed lncRNAs and mRNAs. The “cor. test” function in R language was used for correlation calculation and test, and the “Pearson” method was used to find the Pearson correlation coefficient (*p* < 0.01) between lncRNA and mRNA to find mRNA co-expressed with lncRNA. A Pearson correlation coefficient R > 0.9 or R < −0.9 was statistically significant. The lncRNA/mRNA co-expression network was constructed by Cytoscape software version 3.9.1 (National Institute of General Medical Sciences, Bethesda, MD, USA). Circular points represent mRNA and triangular points lncRNA. Correlated lncRNA and mRNA are connected by straight lines.

### 4.6. Functional Enrichment Analysis

For functional enrichment analysis of lncRNA, their co-expressed mRNA and the “TCGABiolinks” package in R language were used for biological processes (BP) of Gene Ontology (GO) functional enrichment analysis (FDR < 0.01). The biological processes of each lncRNA were screened, and lncRNA associated with autophagy, proliferation, apoptosis, and senescence were selected as candidates for further functional research.

### 4.7. Cell Culture

Mouse hippocampus-derived neuronal HT22 cells (SCC129, Merck KGaA, Darmstadt, Germany) were cultured in MEM medium contained with 10% fetal bovine serum (PM150410B, Pricella, Wuhan, China) and antibiotics (100 U/mL streptomycin, 100 U/mL penicillin) (PB180120, Pricella, Wuhan, China) at 37 °C in a humidity chamber with 5% CO_2_ and 95% air. The cells were sub-cultured or treated with drugs when they grew to 90% confluence.

### 4.8. Construction of shRNA Expression Vector

ShRNA-NC and shRNA-NEAT1 expression vectors were designed and synthesized by Gemma Biology Technology (Shanghai, China). Homology analysis was performed on the BLAST database to ensure the specificity of the shRNA. The constructed shRNA expression vector is an efficient ready-to-use vector, which continuously produces shRNA in cells, to achieve the purpose of lasting inhibition of target Gene expression. The plasmid used to construct the shRNA expression vector is pGPH1/GFP/Neo plasmid, and its specific information is shown in Appendix A. Detailed information of the shRNA-NEAT1 expression vector is listed in Appendix A.

### 4.9. Transfection of shRNA Expression Vector for RNA Interference

We took a 6-well plate as an example for the experiment: cells were incubated with the complete medium on a 6-well plate at the density of 4 × 10^5^ cells/well. When the cells covered up to 70–80%, the medium was changed to 1.5 mL/well MEM (Pricella, Wuhan, China) culture medium without serum and antibiotics. Then, 10 μL NC or shRNA expression vectors were added to 240 μL of MEM without serum and antibiotics. Afterwards, 5 μL X-tremeGENE^TM^ HP DNA Transfection Reagent (Roche, Basel, Switzerland) was added to 245 μL of MEM without serum and antibiotics. After mixing, they were left at room temperature for 5 min. Finally, the above two 250 μL liquids were combined, mixed, left to stand at room temperature for 20 min, and added to a well of the plate. After 48 h of culture in an incubator, the medium was replaced by a fresh complete medium. RNA expression was detected to analyze RNA interference efficiency.

### 4.10. CCK-8 Assay

Cell proliferation and viability were measured by CCK-8 assay. Exponentially growing cells were seeded at a density of 4 × 10^3^ cells per well in 96-well plates and incubated overnight for 24 h at 37 °C in a 5% CO_2_ incubator. Then, the cells were treated with CCK-8 reagent (Yiyuan Biotechnology, Guangzhou, China) and incubated for 2 h at 37 °C. The absolute OD value was measured at a wavelength of 450 nm using a microplate reader, and the relative OD value was calculated to control group. Each group was set up in three wells, and each measurement was repeated at least three times.

### 4.11. Apoptosis-Associated Hoechst Staining

Apoptosis-associated Hoechst staining was performed using a Hoechst 33258 staining kit (Beyotime Biotechnology, Shanghai, China). Briefly, cells were incubated in fixation fluid for 10 min at room temperature. After washing by PBS, the cells were incubated in Hoechst 33258 staining fluid for 5 min at room temperature while protected from light. Then, the cells were washed with PBS and observed under EVOS™ M5000 Imaging System (Thermo Fisher Scientific, Waltham, MA, USA) through the DAPI channel.

### 4.12. Western Blot Analysis

Whole cellular protein was extracted from HT22 cells. First, 10 μg of the protein was separated by 12% SDS-PAGE and transferred onto PVDF (Thermo Fisher Scientific, Waltham, MA, USA) membranes. The membranes were blocked with TBST contained with 5% nonfat milk for 1.5 h at room temperature. Then, the membranes were incubated with rabbit polyclonal antibody SLC38A2 (Affinity, Changzhou, China) and β-actin (Elabscience Biotechnology, Wuhan, China) overnight at 4 °C. After washing by TBST, the membranes were incubated with peroxidase-conjugated goat anti-rabat or goat anti-mouse IgG (Sungene Biotechnology, Tianjin, Chian) for 1 h at room temperature. The signal of the immuno-band was detected by an ECL reagent (Engreen Biosystem, Beijing, China). The relative optical density values of protein bands were quantified with Image J software (version 1.46r, National Institutes of Health, Bethesda, MD, USA) and calculated to the control group.

### 4.13. Immunofluorescence Staining

Cells were washed with PBS and fixed with 4% formaldehyde at 37 °C for 10 min. After washing by PBS, cells were blocked with goat serum at 37 °C for 1 h and then incubated with rabbit polyclonal antibody SLC38A2 overnight at 4 °C. After being washed with PBS, cells were incubated with Fluor594-conjugated goat anti-rabbit IgG (Yeasen, Shanghai, China) for 1 h at room temperature. Following PBS washing, the cells were incubated with DAPI (Yeasen, Shanghai, China) for 10 min at room temperature. Immunofluorescence was visualized and captured under a laser confocal microscope (Nikon, Tokyo, Japan).

### 4.14. qPCR

Total RNA was extracted from cells using Trizol (15596-026, Invitrogen, Carlsbad, CA, USA) according to the manufacturer’s instructions. RNA quantity and quality were measured by NanoDrop ND-1000 (Thermo Fisher Scientific, Waltham, MA, USA). Complementary DNAs (cDNAs) were synthesized using 0.5 mg of total RNA, oligo(dT)12–18 primers, and a ReverTra Ace qPCR RT kit (FSQ-101, Toyobo, Osaka, Japan) following the manufacturer’s protocol. Gene expression was detected by qPCR using the cDNAs, THUNDERBIRD SYBR qPCR mix reagents (QPS-201, Toyobo, Osaka, Japan), and gene-specific oligonucleotide primers (listed in Appendix A) with an ABI 7500 fast qPCR system (Applied Biosystems, Carlsbad, CA, USA). The expression level of β-actin was used to normalize the relative abundance of RNAs. Significance was determined by taking the average of the β-actin-normalized 2^−ΔΔCT^ values.

### 4.15. Statistical Analysis

All data, except pathological findings, are presented as mean ± SEM. The significant differences in the data between the two groups were determined by Student’s *t*-test and among the groups by one-way analysis of variance for equality of variances using SPSS version 17.0 (IBM, Chicago, IL, USA). A *p* value < 0.05 was considered statistically significant.

## 5. Conclusions

In summary, 38 differentially expressed lncRNAs and 1193 differentially expressed mRNAs were predicted to be associated with oxidative injury in mouse neurons. Of the 38 lncRNAs, 3 lncRNAs NEAT1, 1810026B05Rik, and SNHG12 showed relatively high homology between humans and mice and were related to H_2_O_2_-induce injury in HT22 cells. Moreover, melatonin directly upregulated NEAT1, which was predicted to be involved in proliferation, apoptosis, senescence, and autophagy. The knockdown of NEAT1 aggravated H_2_O_2_-induced oxidative injury, inhibited the protective effect of melatonin on the injury, and disturbed the melatonin-regulated levels of *Eif2ak3* and *Slc38a2*. In conclusion, melatonin attenuates H_2_O_2_-induced oxidative injury by upregulating lncRNA NEAT1 in HT22 cells, which is required for melatonin stabilizing the expression levels of *Eif2ak3* and *Slc38a2* for HT22 cell survival. These results indicate that melatonin regulates lncRNAs not only in physiological conditions but also in pathological conditions of oxidative injury, providing a deeper understanding of melatonin’s protective and antioxidant activities in neurons. Our findings could provide fundamental support for future studies into melatonin and lncRNA therapies for hippocampal senescence or other types of neuronal degeneration.

## Figures and Tables

**Figure 1 ijms-23-12891-f001:**
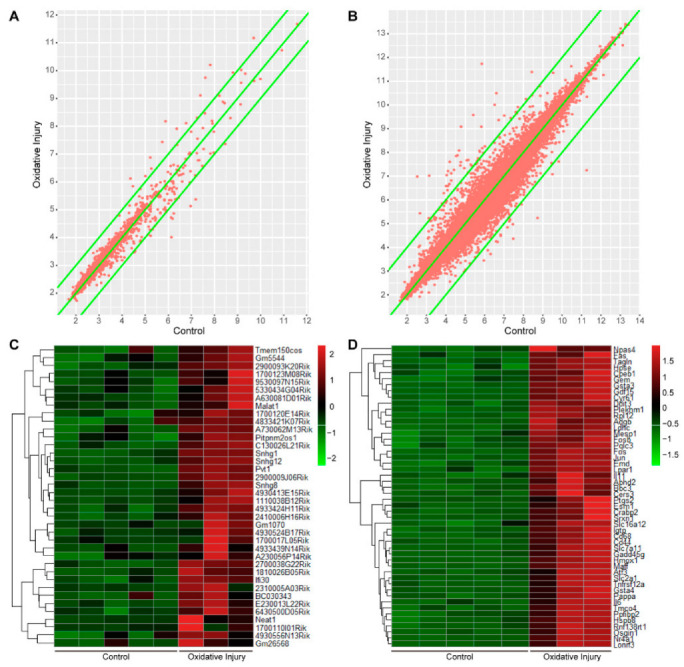
Microarray analysis for identification of differentially expressed lncRNA and mRNA during oxidative injury in mouse neurons. (**A**) The scatter plot of the expression distributions of lncRNA in the oxidative injury group and the control group. (**B**) The scatter plot of the expression distributions of mRNA in the oxidative injury group and the control group. In (**A**,**B**), the horizontal axis represents the value of lncRNA and mRNA in the control group, and the vertical axis represents the value of lncRNA and mRNA in the oxidative injury group. The red dots represent lncRNA or mRNA in each sample. The lncRNA or mRNA distributed near the intermediate line (x = y) represents a similar expression in the two groups. The points outside of the two edges (|y − x| > 1) represent lncRNA or mRNA that are differentially expressed greater than a 2-fold change between the two groups. (**C**) The hierarchical clustering heat map of differentially expressed lncRNA between the oxidative injury group and the control group. (**D**) The hierarchical clustering heat map of differentially expressed mRNA between the oxidative injury group and the control group. In (**C**,**D**), the red color indicates high relative expression, and green indicates low relative expression. N = 5 for the control group, and N = 3 for the oxidative injury group. Con represents the control group.

**Figure 2 ijms-23-12891-f002:**
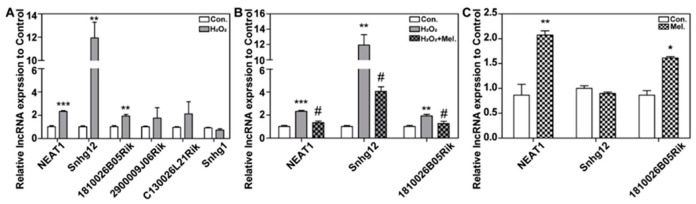
LncRNA nuclear paraspeckle assembly transcript 1 (NEAT1) and 1810026B05Rik were involved in melatonin protecting against H_2_O_2_-induced oxidative injury in HT22 cells. (**A**) HT22 cells were treated with 200 μM H_2_O_2_ for 24 h. The relative expression of lncRNA NEAT1, 1810026B05Rik, 2900009J06Rik, small nucleolar RNA host gene 12 (SNHG12), SNHG1, and C130026L21Rik were detected by quantitative real-time PCR (qPCR). (**B**) HT22 cells were treated with 200 μM H_2_O_2_ or 200 μM H_2_O_2_ + 50 μM melatonin for 24 h. The relative expression of lncRNA NEAT1, SNHG12, and 1810026B05Rik was detected by qPCR. (**C**) HT22 cells were treated with 50 μM Melatonin for 24 h. The relative expression of lncRNA NEAT1, SNHG12, and 1810026B05Rik was detected by qPCR. Results were expressed by the relative content of the Control group. *, ** and *** respectively represents *p* < 0.05, *p* < 0.01 and *p* < 0.001 compared with the Control group. # represents *p* < 0.05 compared with the H_2_O_2_ group.

**Figure 3 ijms-23-12891-f003:**
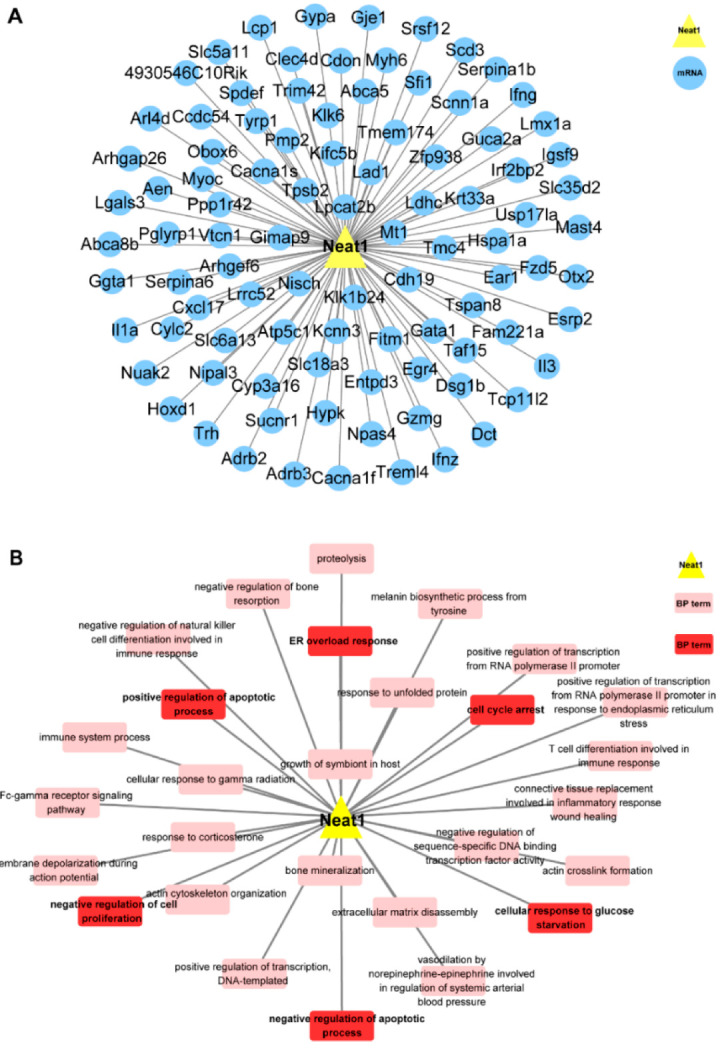
Gene ontology analysis for the potential functionalities of lncRNA NEAT1 during oxidative injury in mouse neurons. (**A**) Co-expression network of LncRNA NEAT1 and its related differentially expressed mRNAs. (**B**) Network of lncRNA NEAT1 and its related biological processes (BP). The yellow triangle represents lncRNA NEAT1, the blue round NEAT1-related mRNA, the pink rectangle biological processes related to NEAT1, and the red rectangle biological processes related to NEAT1 and proliferation, apoptosis, senescence or autophagy.

**Figure 4 ijms-23-12891-f004:**
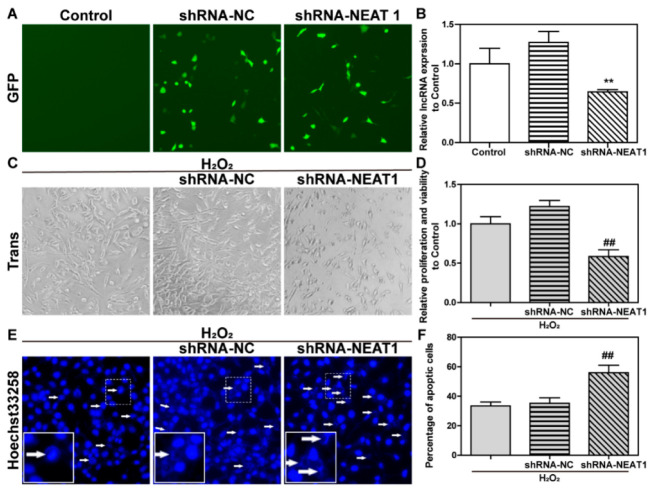
The knockdown of lncRNA NEAT1 aggravated H_2_O_2_-induced oxidative injury in HT22 cells. After HT22 cells were transfected with shRNA-negative control (NC) and shRNA-NEAT expression vectors for 48 h, (**A**) the expression of green fluorescent protein (GFP) carried by the vectors was observed with fluorescence microscopy. The results were obtained at 100× magnification; (**B**) the expression level of NEAT1 was detected by qPCR. Then, HT22 cells were treated with H_2_O_2_ for 24 h following the transfection of vectors. (**C**) The morphological change was observed under phase contrast microscopy. The results were obtained at 100× magnification. (**D**) Relative cellular viability was detected by Cell Counting Kit (CCK)-8. (**E**) Hoechst33258 staining was used to observe apoptosis. The white arrows indicate nuclear fragmentation and nuclear consolidation in HT22 cells. The result was obtained at 200× magnification. (**F**) Statistical results of (**E**). ** represents *p* < 0.01 compared to the shRNA-NC group. ## represents *p* < 0.01 and *p* < 0.001 compared to the H_2_O_2_ + shRNA-NC group.

**Figure 5 ijms-23-12891-f005:**
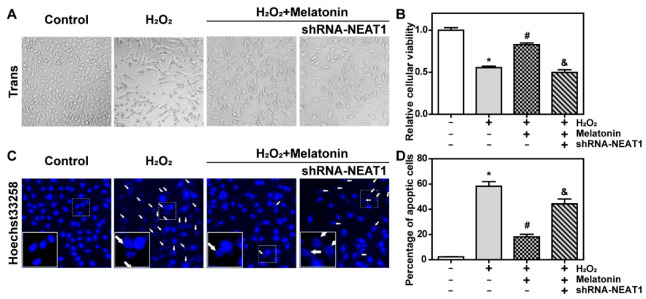
The knockdown of lncRNA NEAT1 inhibited the protective effect of melatonin on H_2_O_2_-induced oxidative injury. HT22 cells were transfected with shRNA-NEAT1 expression vectors for 48 h and then treated with H_2_O_2_ or H_2_O_2_ + melatonin for 24 h. (**A**) Cell morphology was observed by phase contrast microscopy. The result was obtained at 100× magnification. (**B**) Relative cellular viability was detected by CCK-8. (**C**) Apoptosis was observed by Hoechst33258 staining. The white arrows indicate nuclear fragmentation and nuclear consolidation in HT22 cells. The result was obtained at 200× magnification. (**D**) Statistical results of (**C**). * represents *p* < 0.05 compared with the control group, # represents *p* < 0.05 compared with the H_2_O_2_ group, & represents *p* < 0.05 compared with the H_2_O_2_ + melatonin group.

**Figure 6 ijms-23-12891-f006:**
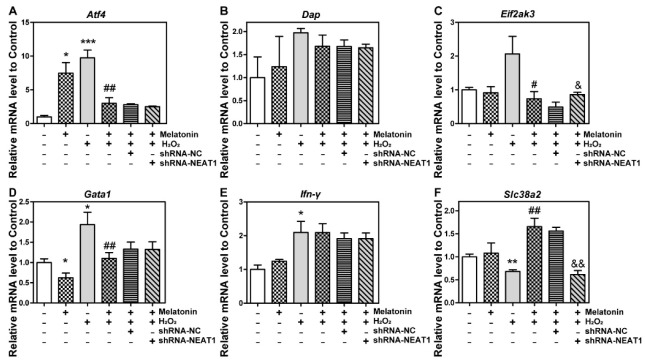
The influence of H_2_O_2_, melatonin, and shRNA-NEAT1 on the expression of NEAT1 co-expressed mRNA. HT22 cells were transfected with or without shRNA-NC or shRNA-NEAT1 vector for 48 h, followed by treatment with melatonin, H_2_O_2_, or H_2_O_2_ + melatonin for 24 h. Then, the mRNA level of *Atf4* (**A**), *Dap* (**B**), *Eif2ak3* (**C**), *Gata1* (**D**), *Ifn-γ* (**E**), and *Slc38a2* (**F**) was detected by qPCR. *, ** and ***, respectively, represent *p* < 0.05, *p* < 0.01 and *p* < 0.001 compared with the control group. # and ##, respectively, represent *p* < 0.05 and *p* < 0.01 compared with the H_2_O_2_ group. & and &&, respectively, represent *p* < 0.05 and *p* < 0.01 compared with the H_2_O_2_ + melatonin + shRNA-NC group.

**Figure 7 ijms-23-12891-f007:**
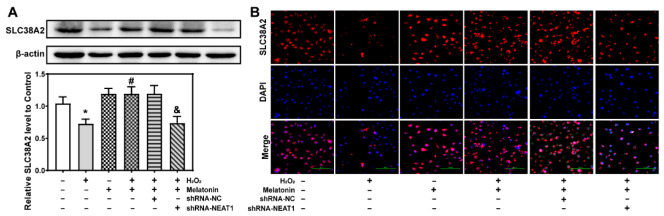
The influence of H_2_O_2_, melatonin, and shRNA-NEAT1 on the expression of sodium-coupled neutral amino acid transporter 2 (SLC38A2). HT22 cells were transfected with or without shRNA-NC or shRNA-NEAT1 vector for 48 h, followed by treatment with melatonin, H_2_O_2,_ or H_2_O_2_ + melatonin for 24 h. (**A**) The expression level of sodium-coupled neutral amino acid transporter 2 (SLC38A2) was detected by Western blot analysis. The relative optical density values of SLC38A2 to β-actin were quantified using Image J software version 1.46r. (**B**) The expression level of SLC38A2 on membranes was detected by immunofluorescence staining. * represents *p* < 0.05 compared with the control group. # represents *p* < 0.05 compared with the H_2_O_2_ group. & represents *p* < 0.05 and *p* < 0.01 compared with the H_2_O_2_ + melatonin + shRNA-NC group.

**Table 1 ijms-23-12891-t001:** Expression profile data of long non-coding RNA (lncRNA) and mRNA.

Data Categories	Data Quantity
Sample	8 (control group 5, oxidative injury group 3)
Original probe	45,057
Specific probe	38,300
lncRNA related probe	1800
mRNA related probe	31,929
Detected lncRNA	1439
Detected mRNA	15,965

**Table 2 ijms-23-12891-t002:** The top 10 differentially expressed lncRNA during oxidative injury in mouse neurons.

Gene Symbol	Seq. Name	Fold Change	*p*-Value	Reg.	RNA Length (b)	Model (Normalized)	Control (Normalized)
C130026L21Rik	BB_632665	7.9	<0.0001	up	2472	7.899769333	0.999999926
SNHG1	BQ_177137	4.783	<0.0001	up	605	4.325219667	0.958542270
SNHG12	BI_180630	5.053	<0.0001	up	935	5.052909467	1.000000068
4930413E15Rik	AK_015127	10.694	<0.0001	up	1290	10.69391797	1.000000032
E230013L22Rik	BB_549061	7.115	<0.0001	up	1856	7.115391	0.999999986
A730062M13Rik	BB_254594	4.611	<0.0001	up	1910	4.611339767	1.000000040
1110038B12Rik	BE_133150	4.261	<0.0001	up	785	4.261468833	1.000000048
1810026B05Rik	BB_327381	3.974	<0.0001	up	5140	3.413172133	0.999999991
2310005A03Rik	AK_009160	8.082	<0.0001	up	1398	8.0818338	0.999999980
1700123M08Rik	BB_408149	5.446	<0.0001	up	770	5.445621267	0.999999948

Seq. name—lncRNA name. Fold change—absolute fold change between the two groups. Reg.—regulation in the oxidative injury models compared to control; “Up” indicates upregulated lncRNA in the oxidative injury models compared with control; Model/Control (normalized)—normalized intensities of each sample (log2-transformed). The list only shows the top 10 of the results of lncRNA with an upregulation in expression in oxidative injury models vs. control.

**Table 3 ijms-23-12891-t003:** The top 10 differentially expressed mRNA with an up or downregulation during oxidative injury in mouse neurons.

Gene Symbol	Seq. Name	Fold Change	*p*-Value	Reg.	RNA Length (b)	Model (Normalized)	Control (Normalized)
Ptgs2	M94967	62.36897551	0.049713300	up	4453	62.368978	1.000000040
Gsta3	AI_172943	15.11052768	0.001807316	up	1457	15.11052724	0.999999971
Cers3	AI_429073	6.985921284	0.026982869	up	3250	6.9859212	0.999999988
Hmox1	NM_010442	73.17679532	0.005781206	up	1569	73.1768	1.000000064
Gdf15	NM_011819	20.12441644	0.006973339	up	1545	20.12441467	0.999999912
Npas4	AV_348246	11.1304136	0.027445049	up	3277	11.13041367	1.000000006
Ptgr1	BC_014865	15.36713818	0.040184910	up	2296	15.36713833	1.000000010
Gem	U10551	5.740063778	0.004308775	up	1994	5.740063767	0.999999998
Cpeb1	NM_007755	5.963823015	0.003356431	up	3111	5.9638228	0.999999964
Tlk2	NM_011903	2.990434232	0.002009714	up	5494	2.990434167	0.999999978
Dclre1b	BB_763339	0.258067778	0.002308249	down	4113	0.259593163	1.005910792
Olig2	AB_038697	0.076296681	0.021070358	down	2437	0.076296676	0.999999928
Pdgfra	AW_537708	0.115138148	0.010354445	down	2876	0.114080087	0.990810510
Hes5	AV_337579	0.118642876	0.041823638	down	1273	0.118642871	0.999999957
Sox8	AV_345303	0.151596202	0.006506986	down	3000	0.151596204	1.000000018
Ascl1	BB_425719	0.137524736	0.009259114	down	2481	0.137524738	1.000000016
Nim1k	BB_359887	0.285656255	0.007794788	down	4103	0.285656243	0.999999960
Mki67	X82786	0.229441042	0.008072360	down	100,061	0.22944107	1.000000120
Homez	AV_298304	0.20154121	0.031646867	down	5609	0.20154121	1.000000000
Bbs12	AI_449447	0.188051542	0.023960328	down	2433	0.188051538	0.999999980

Seq name—mRNA name. Fold change—absolute fold change between the two groups. Reg.—regulation in the oxidative injury models compared to control; “Up” indicates upregulated mRNA in the oxidative injury models compared with control; “Down” indicates downregulated one in the oxidative injury models compared with control. Model/Control (normalized)—normalized intensities of each sample (log2-transformed). The list only shows the top 10 of the results of mRNA with an up or downregulation in expression in the oxidative injury models vs. control.

**Table 4 ijms-23-12891-t004:** Homology analysis of lncRNA between humans and mice.

Gene Symbol	Homology Score	Homology Degree
2900009J06Rik	222	High
1810026B05Rik	286	High
NEAT1	273	High
SNHG12	163	Medium
SNHG1	165	Medium
C130026L21Rik	102	Medium

**Table 5 ijms-23-12891-t005:** Genomic information of the six mouse lncRNAs.

Gene Symbol	Seq. Name	Biotype	Fold Change	*p*-Value	Reg.	RNA Length (b)	Chrom	Strand	txStart	txEnd
2900009J06Rik	BB_663189	Antisense	2.544	<0.0001	up	465	1	−	127,756,757	127,767,978
1810026B05Rik	BB_327381	lincRNA	3.974	<0.0001	up	5140	7	−	73,539,801	73,558,395
NEAT1	AK_018202	lincRNA	7.38	<0.0001	up	20,771	19	−	5,824,708	5,845,478
SNHG12	BI_180630	lincRNA	5.053	<0.0001	up	935	4	+	132,308,678	132,311,024
SNHG1	BQ_177137	lincRNA	4.783	<0.0001	up	605	19	+	8,723,475	8,726,443
C130026L21Rik	BB_632665	lincRNA	7.9	<0.0001	up	2472	5	+	111,581,422	111,587,945

Seq name—mRNA name. Fold change—absolute fold change between the two groups. Reg.—regulation in the neuron injury model compared to control. Chrom—the number of the chromosome where lncRNA locates. Strand—the coordinate on the genome where lncRNA start to transcribe. txStart/txEnd—the coordinate on the genome where lncRNA starts/stops to transcribe.

## Data Availability

Not applicable.

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
