# Peer review of "Melatonin Attenuates H2O2-Induced Oxidative Injury by Upregulating LncRNA NEAT1 in HT22 Hippocampal Cells"

_ijms, 2022, doi:10.3390/ijms232112891_

Round 1

Reviewer 1 Report

An article by Gao et al is devoted to the study of how lncRNA is involved in melatonin protecting the hippocampus from H2O2-induced oxidative injury. In their work, the authors use a bioinformatics approach, as well as methods of cellular and molecular biology. Based on the data obtained, the authors conclude that melatonin attenuates H2O2-induced oxidative injury by up-regulating lncRNA NEAT1, which is essential for melatonin stabilizing the mRNA and protein level of Slc38a2 for the survival of HT22 cells

Comments:

1. The main idea of this work is that lncRNA is the regulatory molecule responsible for a wide range of observed effects. At the same time, the authors do not talk about other intracellular targets of melatonin and H2O2.

2. Why did the authors study the influence of H2O2, melatonin, and shRNA-NEAT1 on the expression of Atf4, Dap, Eif2ak3, 276 Gata1, Ifn-γ, and Slc38a2? The authors should clarify this.

Author Response

Responds to the reviewer’s comments:

Referee #1:

Point 1: The main idea of this work is that lncRNA is the regulatory molecule responsible for a wide range of observed effects. At the same time, the authors do not talk about other intracellular targets of melatonin and H2O2.

Response1: We thank you very much for your precious comment and suggestion. LncRNA is indeed the regulatory molecule responsible for a wide range of observed effects. Because lncRNAs are widely found to regulate multiple diseases, including neuro-oxidative damage. We previously found that melatonin inhibits H2O2-induced oxidative injury through activating autophagy. So, we wanted to see whether and how lncRNA mediates the protective effects of melatonin in neurons.

In this study, we firstly predicted 1,439 lncRNA and 15,965 mRNA associated with oxidative damage by bioinformatics. Then, we found two lncRNA, NEAT1 and 1810026B05Rik, which may be regulated by melatonin and H2O2.

Then in this study, we mainly studied the relationship between NEAT1 and H2O2 and melatonin, and further found their related target Slc38a2. We found that melatonin up-regulated lncRNA NEAT1. Silencing NEAT1 resulted in a variety of effects, including aggravating H2O2-induced oxidative injury, weakening melatonin’s neuron-protecting function, and interfering with Slc38a2 protein and mRNA expression. Therefore, lncRNA NEAT1 is worth further investigation in its functions.

It’s also true that we do not talk about other intracellular targets of melatonin and H2O2. Because, another lncRNA 1810 and other targets related to H2O2 and melatonin are still under investigation, and the results may be published in the future.

Point 2: Why did the authors study the influence of H2O2, melatonin, and shRNA-NEAT1 on the expression of Atf4, Dap, Eif2ak3, Gata1, Ifn-γ, and Slc38a2? The authors should clarify this.

Response2: Thank you very much for your good comments. After we knew that silencing NEAT1 weakens the anti-H2O2 effect of melatonin, we wanted to figure out if it was caused by interference with the NEAT1-related mRNA which may also be regulated by H2O2 and melatonin.

In the analysis of bioinformatics, we screened out 1,439 lncRNA associated with oxidative damage, among which 6 lncRNA (NEAT1, 1810026B05Rik, 2900009J06Rik, SNHG12, SNHG1, and C130026L21Rik) with high and medium homology between human and mouse were screened (Table 4). The 6 lncRNA were further predicted to be closely related to 806 differentially expressed mRNA, which formed 2701 lncRNA-mRNA correlation pairs (Supplementary Figure S2). Among these lncRNA-related mRNA, we selected several mRNAs reported to be related to proliferation, apoptosis, senescence, and autophagy. So, the candidate mRNA, including Atf4, Dap, Eif2ak3, Gata1, Ifn-γ, and Slc38a2, were detected after the treatment of H2O2, melatonin, and shRNA-NEAT1. And the result let us know that Slc38a2 is a target molecular of H2O2, melatonin, and NEAT1.

Special thanks to you for your comments. We tried our best to improve the manuscript and made some changes in the manuscript. These changes will not influence the content and framework of the paper. And here we did not list the changes but marked them in Red under the “Track Changes” function in the revised paper. We appreciate for Editors’ and Reviewers’ warm work earnestly and hope that the correction will meet with approval. Once again, thank you very much for your comments and suggestions.

Reviewer 2 Report

Wide-ranging use of bioinformatics in life science research facilitates hypothesis validation and often leads to unexpected discoveries. In the study by Gao et al. the Authors applied this approach to investigate the mechanism of melatonin-dependent protection of neurons from oxidative stress. Data-mining techniques identified a number of lncRNA, both mouse and human, that could be involved in melatonin pathways. The Authors used in vitro model to validate the bioinformatic results and confirmed the involvement of ncRNA NEAT1 and Slc38a2 in the melatonin -dependent antioxidant protection. The presented study presents a very-well designed, modern approach to identify molecular pathways underlying a physiological process. The text needs to be linguistically uplifted to meet the criteria of the Journal.   

Author Response

Referee #2:

Point 1: Wide-ranging use of bioinformatics in life science research facilitates hypothesis validation and often leads to unexpected discoveries. In the study by Gao et al. the Authors applied this approach to investigate the mechanism of melatonin-dependent protection of neurons from oxidative stress. Data-mining techniques identified a number of lncRNA, both mouse and human, that could be involved in melatonin pathways. The Authors used in vitro model to validate the bioinformatic results and confirmed the involvement of ncRNA NEAT1 and Slc38a2 in the melatonin-dependent antioxidant protection. The presented study presents a very-well designed, modern approach to identify molecular pathways underlying a physiological process. The text needs to be linguistically uplifted to meet the criteria of the Journal. 

Response 1: Thank you very much for your good comments. We apologize for the original writing error. So, we have tried our best to improve the manuscript and made some changes in the manuscript. Therefore, the text has been linguistically uplifted. These changes will not influence the content and framework of the paper. And here we did not list the changes but marked them in Red under the “Track Changes” function in the revised paper.

We appreciate for Editors’ and Reviewers’ warm work earnestly and hope that the correction will meet with approval. Once again, thank you very much for your comments and suggestions.

Round 2

Reviewer 1 Report

The manuscript is appropriately revised and it can be published in the current state.